# A new dataset for measuring the performance of blood vessel segmentation methods under distribution shifts

**Matheus Viana da Silva**[1]*, **Natália de Carvalho Santos**[2], **Julie Ouellette**[3,4], **Baptiste Lacoste**[3,4], **Cesar H. Comin**[1]

1 Department of Computer Science, Federal University of São Carlos, São Carlos, Brazil, 2 São Carlos Institute of Physics, University of São Paulo, São Carlos, Brazil, 3 Department of Cellular and Molecular Medicine, Faculty of Medicine, University of Ottawa, Ottawa, Canada, 4 Neuroscience Program, Ottawa Hospital Research Institute, Ottawa, Canada

* mvdscc@gmail.com

**Data availability statement:** The VessMAP dataset is available at

## Abstract

Creating a dataset for training supervised machine learning algorithms can be a demanding task. This is especially true for blood vessel segmentation since one or more specialists are usually required for image annotation, and creating ground truth labels for just a single image can take up to several hours. In addition, it is paramount that the annotated samples represent well the different conditions that might affect the imaged tissues as well as possible changes in the image acquisition process. This can only be achieved by considering samples that are typical in the dataset as well as atypical, or even outlier, samples. We introduce VessMAP, an annotated and highly heterogeneous blood vessel segmentation dataset acquired by carefully sampling relevant images from a large non-annotated dataset containing fluorescence microscopy images. Each image of the dataset contains metadata information regarding the contrast, amount of noise, density, and intensity variability of the vessels. Prototypical and atypical samples were carefully selected from the base dataset using the available metadata information, thus defining an assorted set of images that can be used for measuring the performance of segmentation algorithms on samples that are highly distinct from each other. We show that datasets traditionally used for developing new blood vessel segmentation algorithms tend to have low heterogeneity. Thus, neural networks trained on as few as four samples can generalize well to all other samples. In contrast, the training samples used for the VessMAP dataset can be critical to the generalization capability of a neural network. For instance, training on samples with good contrast leads to models with poor inference quality. Interestingly, while some training sets lead to Dice scores as low as 0.59, a careful selection of the training samples results in a Dice score of 0.85. Thus, the VessMAP dataset can be used for the development of new active learning methods for selecting relevant samples for manual annotation as well as for analyzing the robustness of segmentation models to distribution shifts of the data.

https://zenodo.org/records/10045265. The code for the sampling methodology and the experiments with neural networks is available at https://github.com/matheus-viana/vessmap.

**Funding:** Cesar H. Comin thanks FAPESP (grant no. 21/12354-8) for financial support. M. V. da Silva thanks FAPESP (grant no. 23/03975-4), Google's Latin America Research Awards (LARA 2021), and the Google PhD Fellowship Program for financial support. The authors acknowledge the support of the Government of Canada's New Frontiers in Research Fund (NFRF) (NFRFE-2019-00641).

**Competing interests:** The authors declare that they have no known competing financial interests or personal relationships that could have appeared to influence the work reported in this paper.

# 1 Introduction

The performance of neural networks has dominantly been measured using metrics such as classification or segmentation accuracy, precision, recall, and the area under the ROC curve. However, recent studies have shown the dangers of only considering such globally-averaged metrics [1–3] that provide only an aggregated, summarized, view of the performance of machine learning algorithms on datasets with sometimes millions of images. Such an approach may hide important biases of the model [3]. For instance, for medical images, a 95% accuracy is usually considered a good performance. But what about the remaining 5%? It is usually unrealistic to expect models to reach 100% accuracy, but the samples that are not correctly processed by a neural network may hide important biases of the model. These concerns led to the definition of new approaches and metrics that can aid the interpretation of black box models [4].

For medical image segmentation, the detection of relevant structures is usually only the first step of a more elaborate procedure for measuring relevant properties such as size [5], regularity [6], length [7,8], and curvature [8,9] of the imaged structures. Therefore, systematic segmentation mistakes might lead to undetected errors when characterizing samples for clinical diagnoses [10] and research purposes [1]. An important cause of such systematic errors can be the presence of samples with characteristics that occur with low frequency in a dataset. This can happen due to additional, unexpected, noise during image acquisition, variations in tissue staining, image artifacts, or even the presence of structures that are anatomically different than what was expected. Assuming for illustration purposes that the data is normally distributed, a machine learning model having good performance around the peak of the distribution will tend to have good average accuracy measured for the whole dataset, even if it cannot correctly classify or segment images that are around the tail of the distribution [11], which might be important for downstream analyses.

Segmenting the vasculature in tissue samples tends to be particularly challenging since the appearance of blood vessels can change significantly depending on tissue preparation and imaging protocols. In addition, in most cases, the global shape of the vasculature can be very different among the samples. We argue that blood vessel segmentation methodologies should have good performance, or even be directly optimized, on both prototypical and atypical samples. This focus can lead to models that are more robust to samples located in a sparsely populated region of the feature space of the dataset. In addition, it might also lead to models that generalize better to out-of-distribution samples as well as to new datasets. With these aspects in mind, we create a new dataset which we call the *Feature-Mapped Cortex Vasculature Dataset* (VessMAP). The dataset is designed to be as heterogeneous as possible by including samples having very different characteristics from each other. To this end, we use a simple and intuitive sampling methodology to select a subset of 100 images from a non-annotated base dataset containing 18279 image patches. The selected samples were then manually annotated with pixel-wise accuracy.

The dataset allows the creation of training and validation splits with images having different characteristics, such as contrast and blood vessel density. We show that different splits of the VessMAP dataset lead to very different training and validation results. As illustrated in Fig 1, the performance on the validation set can be dissimilar depending on the samples used for training a neural network. Thus, we expect the dataset to be useful for the development of new segmentation algorithms that are robust under distribution shifts of the data as well as for the validation of novel few-shot and active learning approaches.

(a) (b) (c) (d)

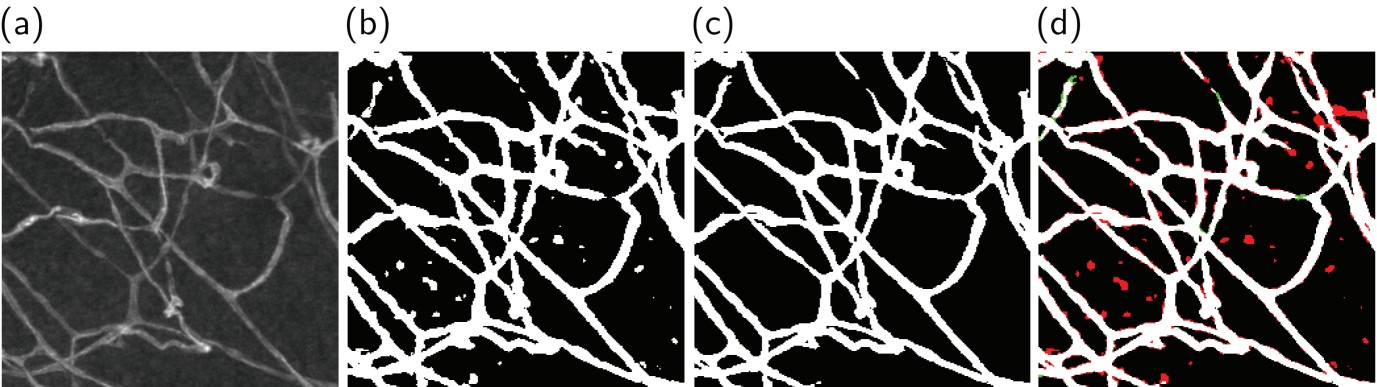

**Fig 1. Illustration of the heterogeneity of the VessMAP dataset.** Two segmentations generated by a neural network trained on different splits of the dataset are shown. (a) Original sample. (b) A network trained on a split of the dataset produces several false positives and misses some blood vessels. (c) The same network trained on a different split generates a more accurate segmentation. (d) Comparison between the two segmentations. Vessel pixels predicted in (b) but not in (c) are shown in red, and pixels in green were predicted as vessels in (c) but not in (b). Since the images in the VessMAP dataset have distinct characteristics, the training set needs to be carefully selected in order to avoid spurious results on the validation set.

The main contributions of this work are listed as follows:

- A new dataset, VessMAP, is made available to aid the development of blood vessel segmentation algorithms;
- It is shown that VessMAP has high variability compared to other popular datasets in the literature, leading to respective large variations of performance when training data is scarce;
- Specific splits of the dataset are provided for testing techniques that improve the generalizability of blood vessel segmentation algorithms.

## 2 Related works

Table 1 shows a summary of the main blood vessel datasets used in the literature as well as some recently published datasets. Most of the datasets have images from the retina. Few datasets are associated with microscopy images. More importantly, to our knowledge, none of the datasets were specifically designed to maximize the diversity of the samples. The diversity on some datasets tends to come as a proxy from the inclusion of healthy and abnormal tissue. For instance, samples in the DRIVE dataset contain diabetic retinopathy, which generates abnormal characteristics in blood vessels and image artifacts such as exudates that are not related to blood vessels. Still, most blood vessels tend to have a well-defined geometry and texture in all samples of the dataset. Thus, it becomes a simple task for a segmentation algorithm to generalize to new unseen samples from the same dataset. It is not surprising that many methods can reach an accuracy larger than 0.94 on the DRIVE dataset [12].

Regarding microscopy images, all datasets found by our survey include very few samples. Usually, very large 3D volumes are annotated in a semi-supervised fashion. They contain large amounts of vessels, but represent a single individual and image acquisition procedure. Therefore, most vessels have similar appearance and it becomes difficult to measure the generalization capability of segmentation methods. With these limitations in mind, we created a dataset that was specifically designed to include blood vessels having very different characteristics.

The creation of the dataset involved the application of a method for selecting relevant samples for annotation. A concept that is similar to the used methodology is the so-called *coreset* [48]. The aim of a coreset is to select a subset of samples that can optimally represent the

**Table 1. Summary of important blood vessel datasets on the literature.**

| Dataset | Anatomical region | Imaging technique |
|---|---|---|
| DRIVE [13] | retina | color fundus photography |
| STARE [14] | | |
| CHASEDB1 [15] | | |
| HRF [16] | | |
| INSPIRE-AVR [17] | | |
| IMAGERET [18,19] | | |
| MESSIDOR [20] | | |
| VICAVR [21] | | |
| ROC [22] | | |
| DRIONS DB [23] | | |
| DR HAGIS [24] | | |
| RET-TORT [25] | | |
| WIDE [26] | | |
| VAMPIRE [27] | | ultra-wide field-of-view fluorescein angiogram |
| IOSTAR [28] | | scanning laser ophthalmoscopy |
| RC-SLO [28] | | |
| Vascular Model Repository [29] | aorta; cerebral, coronary, aortofemoral and pulmonary arteries | CTA and MRA |
| VESSEL12 [30] | human lung | computed tomography |
| 3D-IRCADb-01 [31] | human liver | |
| ASOCA [32] | coronary arteries | cardiac CTA |
| Vascular Synthesizer[33] | synthetic vessels | — |
| VesSAP [34] | mouse brain vasculature | 3D light-sheet microscopy |
| TubeMap [35] | | |
| Di Diovanna et al. [36] | | |
| BvEM [37] | | volume electron microscopy |
| OCTA [38] | | optical coherence microscopy |
| DeepVess [39] | | two-photon microscopy |
| MiniVess [40] | | |
| VesselExpress [41] | mouse brain, heart, and bladder vasculature | 3D light-sheet microscopy |
| SMILE-UHURA [42] | human brain | MRA |
| TopCoW [43] | | MRA and $\mu$CTA |
| DeepVesselNet [44] | human and rat brain | |
| MSD8 [45] | human liver | computed tomography |
| HR-Kidney [46] | mouse kidney | X-ray |
| HiP-CT [47] | human kidney | computed tomography |

Note: *CTA: Computed tomography angiography. MRA: Magnetic resonance angiography. $\mu$CTA: Micro-computed tomography angiography

whole dataset. Many different methodologies and criteria were developed for defining relevant coresets [49–51]. Indeed, the subset defined by our methodology can be associated with a coreset, but in our case, the aim of the methodology and the approach used differs markedly from the usual definition of a coreset. The aim of our methodology is not focused on accurately representing the underlying distribution of the data or preserving the accuracy of a machine learning algorithm, but on providing a relevant dataset for training machine learning algorithms while avoiding the underrepresentation of atypical samples. In addition, many coreset methodologies use a surrogate neural network to estimate latent features or to estimate a degree of uncertainty about each sample, while our methodology is more general in the sense that any set of features obtained from the samples can be used. Furthermore, many related studies consider a similarity metric for selecting relevant samples [49,50], which is a degenerate metric and, therefore cannot provide a full representation of the data distribution.

# 3 Materials and methods

In the following, we describe the methodology used for creating the VessMAP dataset. The methodology is illustrated in Fig 2 and can be divided into three steps: (a) acquisition of the base data from different experiments; (b) characterization of the base dataset according to important morphometry features; (c) selection of samples that uniformly covers the mapped feature space. The base non-annotated dataset used for selecting relevant samples for VessMAP is described in Sect 3.1. The sampling methodology used for selecting appropriate samples from the base dataset is described in Sect 3.2.

## 3.1 Blood vessel microscopy base images

We start from a collection of 2637 confocal microscopy images of mouse brain vasculature. The images were acquired under different experimental conditions in different works published in the literature [52–54]. Conditions include control animals, animals that have suffered a deletion of chromosome 16p11.12, animals that have experienced sense deprivation or sense hyperarousal, samples from stroke regions, and also from different stages of mouse development. The images have sizes from 1376 × 1104 to 2499 × 2005 pixels, totaling around 3.8 GB of data.

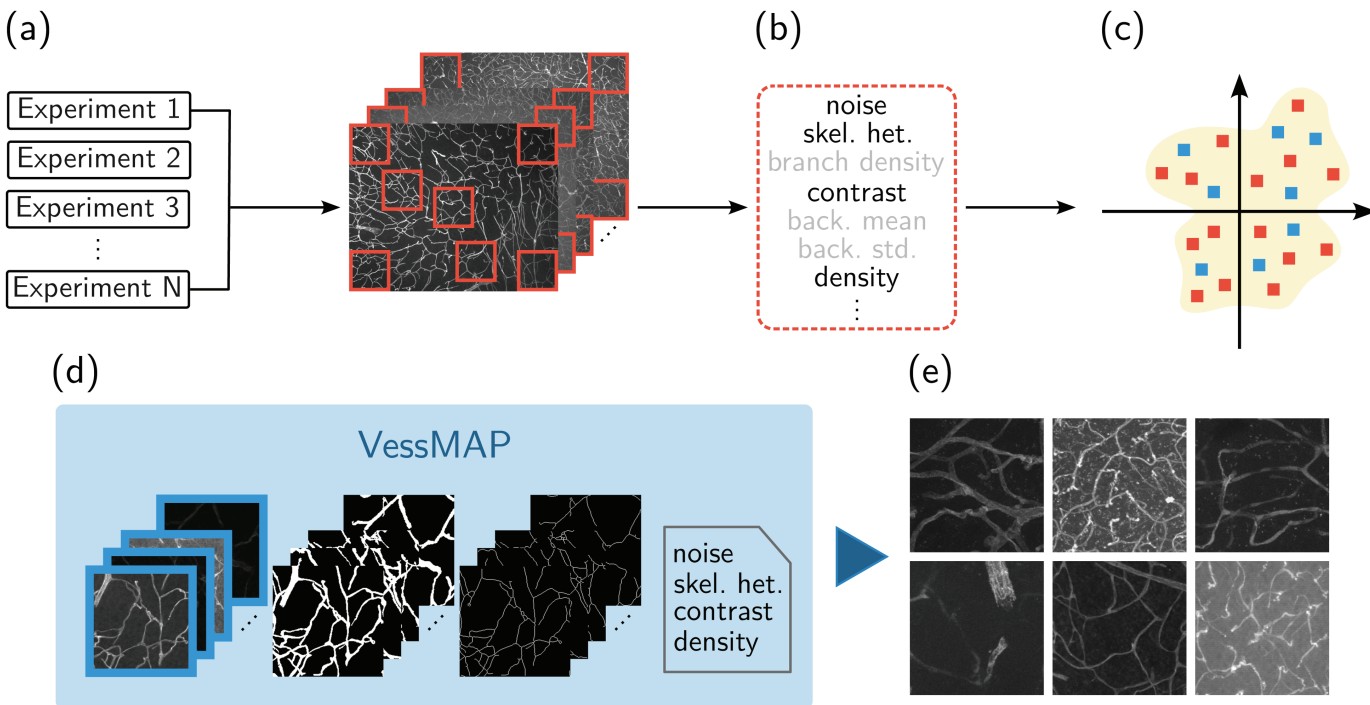

**Fig 2. A flowchart that illustrates the methodology used to create VessMAP.** (a) A large image dataset from many experiments regarding the morphometry of the cortex vasculature of mice is used as the base dataset. (b) A set of random windows is extracted from the base dataset and relevant morphometry features are calculated. (c) The most descriptive features are used to project the samples into a 4-dimensional space and a sampling methodology is applied to select 100 images that uniformly cover this space. (d) The set of 100 images, together with respective manual annotations of the blood vessels, their medial axes, and metadata containing the features of each image, define the VessMAP dataset. (e) Some samples of the dataset are shown. Large differences in vessel appearance can be observed. *skel. het.: skeleton heterogeneity, back. mean.: background mean, back. std.: background standard deviation.

The dataset is interesting because it has a considerable variety of characteristics of blood vessels. In addition, the images represent samples obtained from hundreds of different animals and experimental conditions. This makes it an excellent dataset for training machine learning algorithms for blood vessel segmentation. However, training supervised algorithms requires the manual annotation of the blood vessels.

After annotating a few samples, we estimated that each image in the dataset takes roughly 12 hours to fully annotate. Therefore, it is unfeasible to annotate the whole dataset, and a subset of samples needs to be selected. Our objective was to select a diverse set of samples containing both prototypical and atypical samples, so that it would be possible to create useful training and validation splits for quantifying the performance of segmentation algorithms under challenging distribution shifts between the splits. To this end, a sampling methodology was developed to select appropriate samples.

## 3.2 Sampling methodology

Each image in the base dataset may include illumination inhomogeneities, changes in contrast, different levels of noise, as well as blood vessels having distinct characteristics (e.g., caliber, tortuosity, etc). Thus, from the original dataset, we generated a new set of images, each having a size of $256 \times 256$ pixels. These smaller images were generated by extracting $256 \times 256$ patches from the original images. As shown in Fig 3, seven regions were extracted from each image. The seven regions were extracted in key areas of each image, with four windows in each of the corners of the image, a central window, and two windows at random positions. The latter two may overlap with the other windows. Windows that did not contain a satisfactory number of blood vessel segments were removed. The total size of the resulting dataset is 18279 images. This new dataset was used in the remainder of the sampling procedure.

The methodology developed to sample relevant images has three steps: (1) dataset mapping to a feature space, (2) generation of a discrete representation of the feature space, and (3)

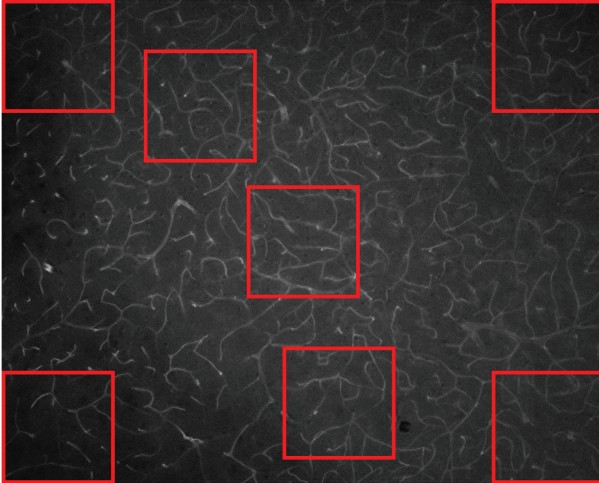

**Fig 3. An example of seven regions extracted from a single sample.** Confocal microscopy images can present illumination differences between the center and the borders of the image. The four corners, along with the central region, can capture most of the illumination inhomogeneities that may occur due to uneven illumination of the samples. Besides these five regions, two additional random regions are also drawn for each image.

selection of points from the feature space representation. We explain each of these steps in the following sections.

**3.2.1 Dataset mapping.** We represent the dataset as $D = \{\delta_1, \delta_2, \ldots, \delta_n\}$ where $n$ is the number of samples. Given a function $f: \delta_i \to \vec{p}_i$ that maps a sample $\delta_i$ to a vector $\vec{p}_i$ with dimension $d$, the dataset is mapped to a feature space as a $n \times d$ matrix, which we call $D_{\text{mapped}}$. The function $f$ represents a set of characteristics measured from the samples. Each line of matrix $D_{\text{mapped}}$ therefore represents the features of a sample $f(\delta_i)$.

Given that the images from our base dataset were used in previous works, each sample has a respective segmentation that was obtained using a semi-supervised methodology. This methodology is based on the adaptive thresholding of the original images, where the threshold was selected manually for each image. The full details of the segmentation procedure are described in [8]. Using the semi-supervised segmentation, the following features were used to characterize the samples: blood vessel contrast, level of Gaussian noise, blood vessel density, and medial line heterogeneity.

The blood vessel contrast is related to the average difference in intensity between the vessels and the background of the image. The greater the contrast, the easier it is to detect the vessels. It can be measured using the original image of the vessels and the respective semi-supervised segmentation containing the pixels belonging to the vessels. The contrast was calculated as

$$C = \frac{\bar{I}_v}{\bar{I}_f}, \tag{1}$$

where $\bar{I}_v$ and $\bar{I}_f$ are the mean intensities of, respectively, the pixels belonging to the blood vessels and the background of the image.

The signal-to-noise level of the images can be estimated in different ways. We investigated different definitions and used the method that was the most compatible with a visual inspection of the images. The method proposed in [55] was used. It assumes a noise with normal distribution and uses wavelets to identify the most likely standard deviation of the noise component. To prevent the method from capturing vessel variation, only the background of the image was used for the estimation.

Blood vessel density is defined as the total length of blood vessels in an image divided by the image area. To do this, we first applied a skeletonization algorithm to extract the medial lines of the vessels [56]. The total length of vessels was then calculated as the sum of the arc-lengths of all vessel segments.

The last metric, which we call medial line heterogeneity, measures the illumination changes in the vessel lumen. To calculate this metric, we first blurred the image using a Gaussian filter with unit standard deviation to remove extreme values. The medial line heterogeneity was then calculated as the standard deviation of the pixel values along the medial lines of this blurred image. The medial lines considered are the same ones used for the blood vessel density metric.

We observed that the medial line heterogeneity tended to be correlated with the average intensity of the blood vessels. In order to remove this dependency, the medial line heterogeneity, as well as the average intensity of the medial lines, were calculated for all images in the dataset. Then, a straight line fit $h_m = a * m + b$ was applied to the calculated values, where $m$ is the average intensity and $h_m$ is the expected medial line heterogeneity associated with $m$. Next, a normalized medial line heterogeneity was defined as $\tilde{h} = h - h_m$, where $h$ is the medial line heterogeneity calculated for an image.

These specific features were used because they can significantly impact the quality of the morphometry assessment of the cortex vasculature. For example, images with low contrast

and/or high noise levels are expected to be more challenging to be accurately segmented. Conversely, images with a larger amount of blood vessels and high medial line heterogeneity provide intricate topology and texture to segmentation algorithms. Other features, such as blood vessel tortuosity, the density of branching points (bifurcations), and additional noise estimators, were considered. However, we disregarded strongly correlated features for our final data selection. The four remaining metrics mapped the base dataset to a 4-D feature space. As mentioned before, the dataset contains 18279 images. Hence, the whole dataset was mapped to a matrix $D_{\text{mapped}}$ having size $18279 \times 4$.

**3.2.2 Feature space discretization.** A regular grid was defined in the 4-D feature space, and each data point was mapped to the nearest point in this grid. Fig 4 illustrates this procedure. For creating the grid, it is useful to first normalize the values of $D_{\text{mapped}}$ to remove differences in the scale of the features. Thus, each feature was normalized to have zero mean and unit variance. Then, the discretization was done by defining a scale $\nu$ that sets the size of each grid cell, and calculating

$$D_{\text{grid}} = \left\lfloor \frac{D_{\text{mapped}}}{\nu} \right\rfloor, \tag{2}$$

where $\lfloor . \rfloor$ represents the floor function. As shown in Fig 4c, this operation ensures that each value of $D_{\text{grid}}$ lies within a regular grid. Note that, as a consequence of undersampling, we expect multiple data points to fall in the same grid position; this is one of the key properties of the method that will allow a uniform sampling of the data. A value of $\nu = 10$ was used since we observed that it provided a good balance between grid sparsity and variability.

After the feature space discretization, we generated a sparse set of points representing an estimation of the possible values that can be obtained in the feature space. We call this set the *sampling set* of the feature space. This procedure works as follows. A 4-dimensional discrete hypersphere $S$ with radius $r$ (in grid units) centered on each data point is defined. This hypersphere is translated to each data point position. The union of the calculated hypersphere positions of all points defines the sampling set $D_{\text{sset}}$. The general appearance of $D_{\text{sset}}$ is depicted by the blue points of Fig 5. The hypersphere radius used for creating the VessMAP dataset was $r = 4$.

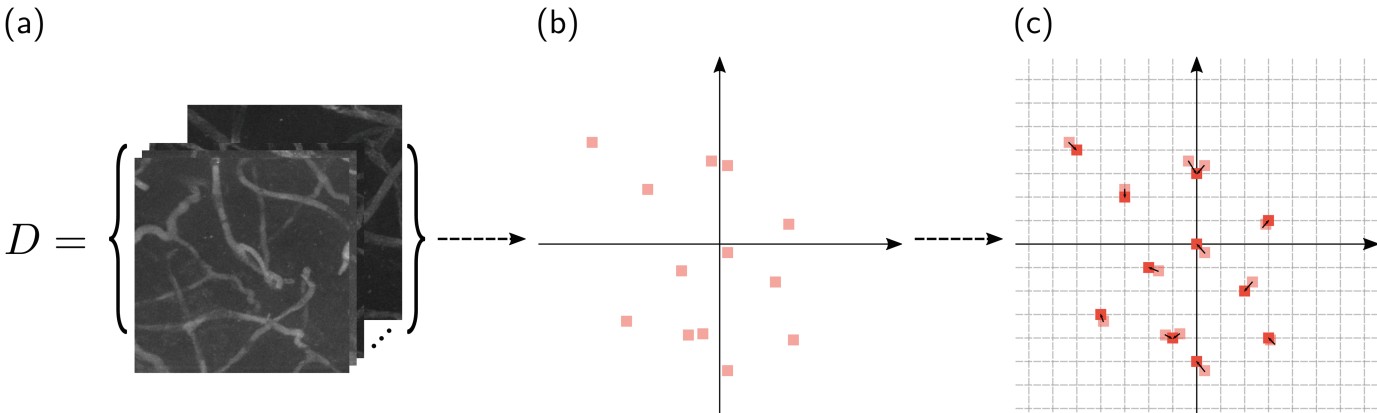

**Fig 4. Representation of the mapping procedure applied to a set $D$ of samples, followed by the feature space discretization.** (a) Set $D$ contains blood vessel images. (b) Each image of $D$ is mapped to a 4-D position in the new feature space. Here, the space is represented in 2-D for ease of visualization. (c) The mapped points (light-red points) are moved to a new position (red points) within a regular grid defined by Eq 2.

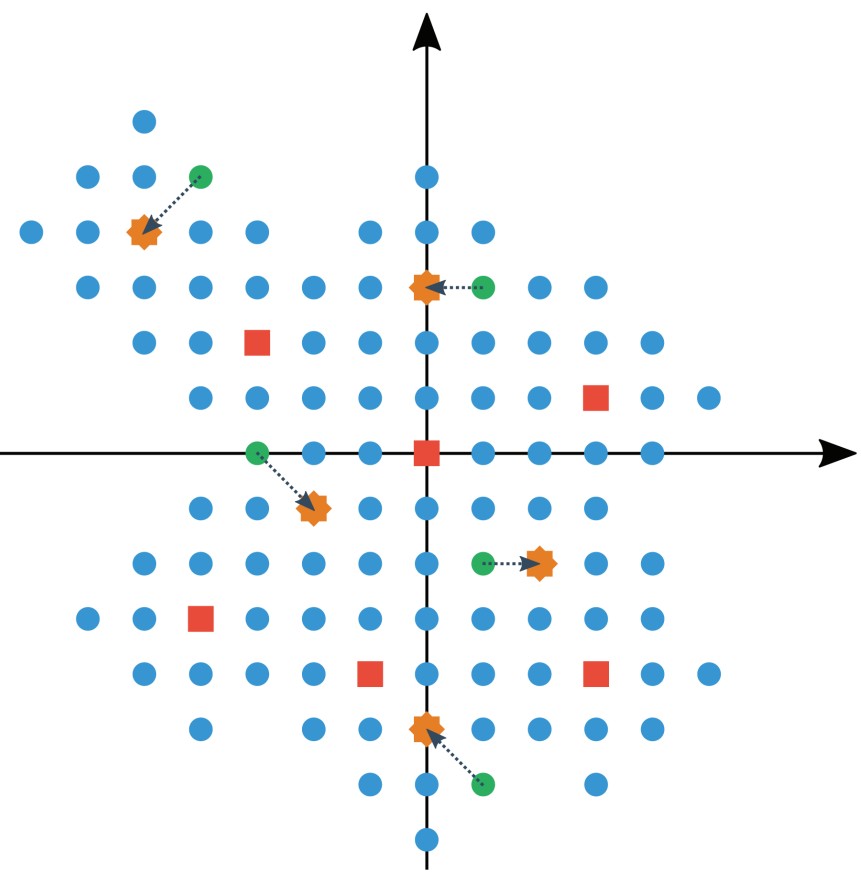

**Fig 5. Illustration of the proposed sampling protocol.** $k$ random points (green dots) are drawn from the sampling set $D_{\text{sset}}$ (blue dots). The subset of sampled data points is defined by the data points that are closest to each drawn point (orange stars). Red squares represent the remaining data points that were not selected.

**3.2.3 Uniform selection of points.** The final step of the method is to select the samples to be manually annotated. The samples are selected by first drawing a set of points from the sampling set $D_{\text{sset}}$. As illustrated in Fig 5, we draw from $D_{\text{sset}}$ $k$ points with uniform probability (green dots in Fig 5). For each point drawn, the closest data sample is identified using the Euclidean distance. If the same data sample is obtained more than once, a new point is drawn from $D_{\text{sset}}$ until $k$ unique data samples are obtained. The final set of data samples (orange stars in Fig 5) is represented as $D_{\text{sampled}}$.

A uniform sampling of $D_{\text{sset}}$ allows the selection of prototypical and atypical samples from the dataset with equal probability. Nevertheless, a single realization of the sampling may lead to distortions, such as the selection of many samples at similar regions of the space or the creation of large regions with no samples selected. This is due to random fluctuations in the sampling process. To amend this, we define a metric called Farthest Unselected Point (FUS) that punishes sampled subsets with large gaps between the selected points.

Let $D_{\text{sampled}}$ be the set of sampled data points from $D_{\text{grid}}$, and $\neg D_{\text{sampled}}$ the set of points from $D_{\text{grid}}$ that were not selected in the sampled subset. For each data point in $\neg D_{\text{sampled}}$, the Euclidean distance to the closest point in $D_{\text{sampled}}$ is obtained. The FUS metric is defined as the largest calculated distance among all points in $\neg D_{\text{sampled}}$. Sampled subsets leading to low values of the FUS metric should be preferred since it avoids the creation of large regions of the

feature space with no samples. In our experiments, we found that minimizing FUS for 1000 different subsets covered a good amount of subset possibilities.

We decided to select $k = 100$ images for annotation. Also, to avoid data leakage, an additional restriction that prevented the selection of samples from the same image was used.

## 4 Results

### 4.1 Dataset heterogeneity

The sampling approach used to generate the VessMAP dataset should lead to a heterogeneous set of samples. It is difficult to properly measure the heterogeneity of the dataset because it would involve the estimation of the probability density function of the original data, which is not a trivial task and can be influenced by the choice of parameter values. However, it is clear that the method should naturally lead to a uniform selection of the samples. This is so because the set $D_{grid}$ (defined in Sect 3.2.2) represents an estimation of the domain of the probability density function of the data, and this domain is sampled uniformly.

One approach to illustrate the characteristics of the sampled images is displayed in Fig 6, which shows histograms of the four considered features for both the full dataset and the sampled subset. The histograms of individual features are not expected to be uniform since they represent a projection of the original data into one dimension. Still, it can be seen that the histograms of the sampled set tend to represent a slightly flattened version of the histograms

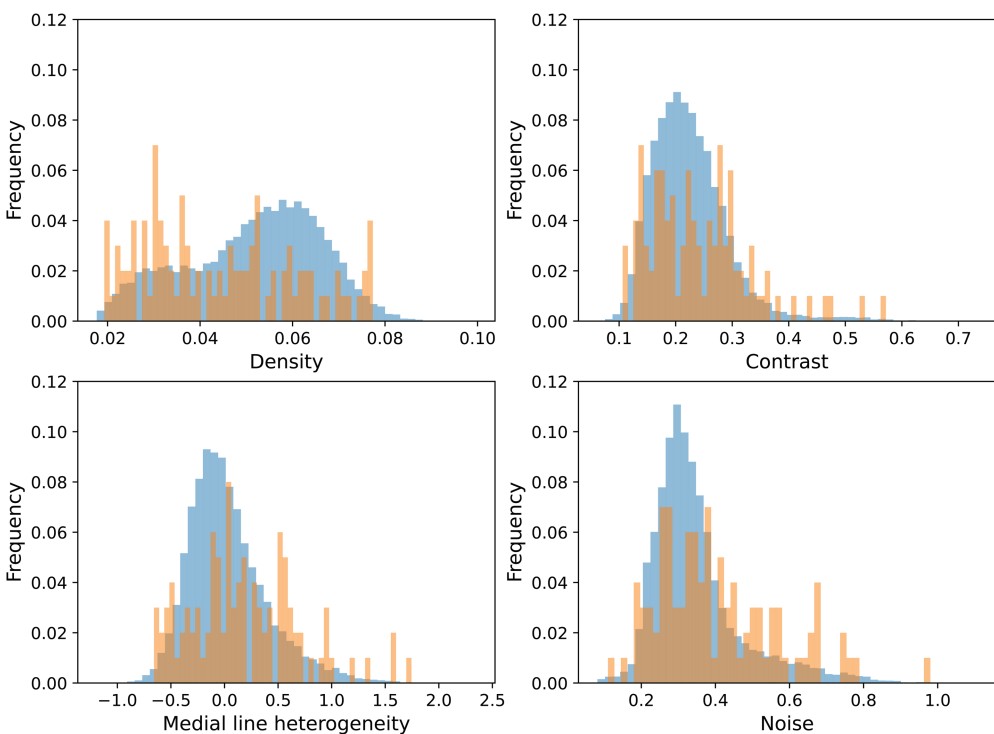

**Fig 6. Histograms of the four features calculated from the base dataset and the sampled subset.** Blue bars correspond to the distribution of each metric of the base dataset of cortex images. Orange bars correspond to the distribution of the sampled subset. Note that the frequencies were normalized by their sum so the y-axis matches for all plots.

of the original data, indicating that a larger priority is being given to atypical samples when compared to the original distribution.

A more robust way of visually checking the sampled subset is to visualize the data using Principal Component Analysis (PCA). Using PCA, the original 4-D data can be projected into 2-D with optimal preservation of the variance [57]. Fig 7 shows the PCA projection of the data. The four plots included in the figure represent the same projection, but the points are colored according to the different features used to characterize the images. The selected samples are shown in red. It can be noticed that the sampling methodology selects a subset of images that uniformly covers the distribution of the data. Furthermore, as also suggested by the histograms in Fig 6, the sampling was capable of covering the full range of values of every considered feature.

The subset of images selected by the method (the VessMAP dataset) is shown in Fig 8. The subset indeed contains a heterogeneous set of images covering many different values of

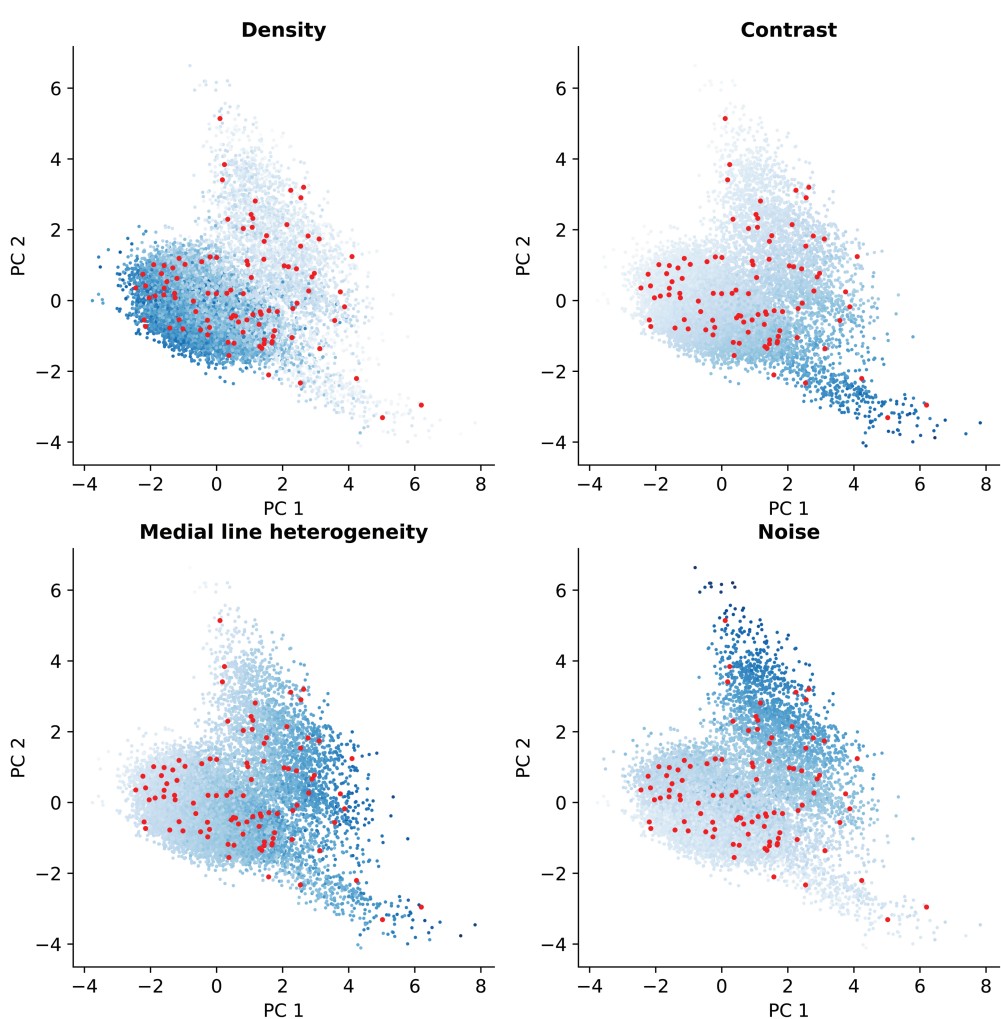

**Fig 7. PCA of the blood vessel dataset.** Red points correspond to the sampled subset obtained by the sampling methodology. Blue points correspond to unselected points from the original dataset, with their lightness representing the value of the four original metrics: vessel density, contrast, medial line heterogeneity, and image noise. Darker blues correspond to larger values of the corresponding metric.

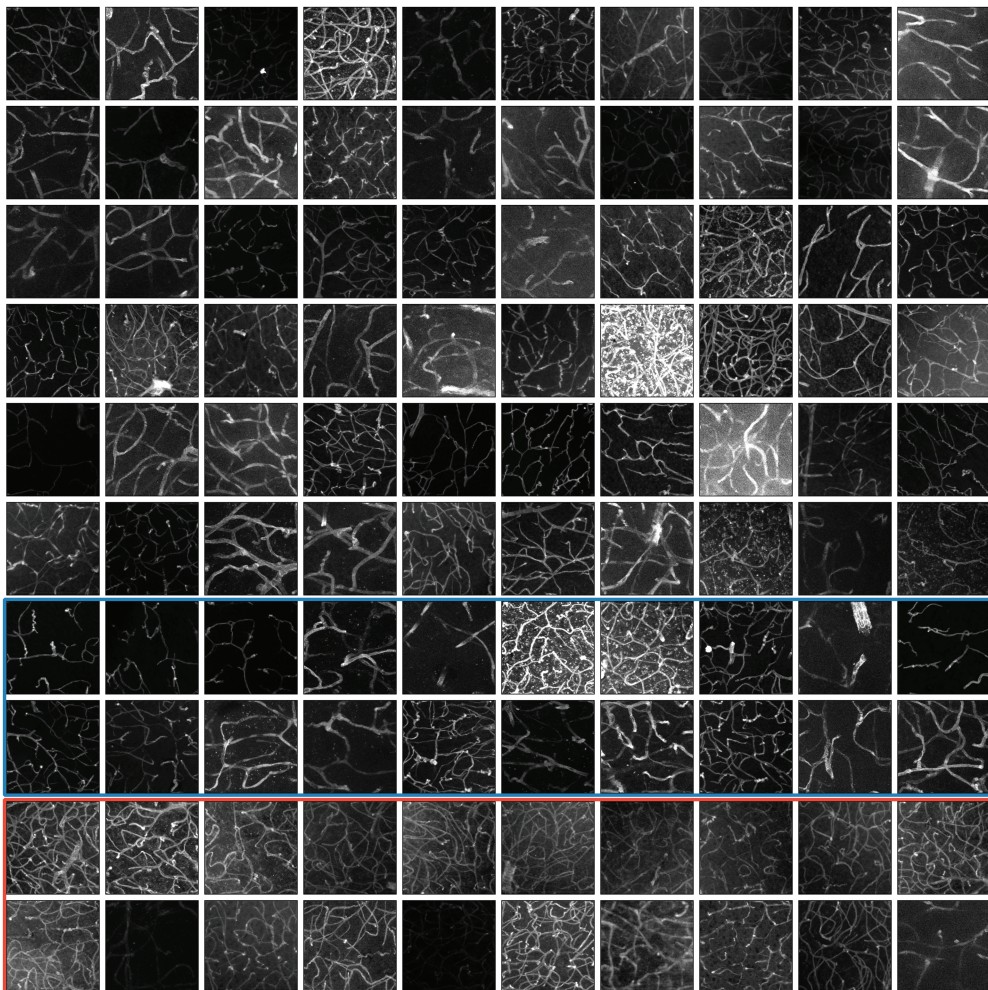

**Fig 8. The VessMAP dataset.** All 100 samples of the dataset are shown in the figure. The images cover a wide range of values in the feature space defined by our four features. Contrast variation and vessel density are the easier features to visually verify. The medial line heterogeneity can be verified by noticing the brightness changes along blood vessels. Gaussian noise level is harder to verify visually, but pronounced noise can be observed on some of the brighter images. Images inside the blue and red rectangles define, respectively, the training and validation sets for some of the experiments described in Sect 4.2.

the considered features (e.g., low contrast, high vessel density, etc). For instance, some of the samples in the dataset come from animals who suffered hemorrhagic strokes. These samples are very different from the typical samples contained in the base dataset, and they would be largely underrepresented if a sampling following the data distribution was performed.

We manually labeled each of the 100 images and made the dataset publicly available [58]. To account for inter-annotator variability, 20 samples were labeled by two annotators. The Dice similarity score between the two annotators is 0.8780. We identified that most disagreement between annotators lies in delineating the vessel borders, resulting in mildly different blood vessel calibers. With that in mind, we also calculated the centerline Dice (clDice) [1] between both annotations, which provides a metric of how well the annotators agreed about the topology of the blood vessels. A clDice of 0.9556 was obtained, indicating a good agreement between annotators regarding the preservation of continuities and bifurcations.

As a comparison, the annotations of the DRIVE dataset's (one of the most used blood vessel segmentation datasets) test set have a Dice similarity of 0.7881 and a clDice of 0.7634.

The VessMAP repository includes manually annotated binary labels, their skeletons (calculated by the Palágyi-Kuba algorithm [56]), and the metrics for each sample (as described in Sect 3.2.1)—which were calculated using the manual annotations. We verified that the metrics calculated from the manual annotations have a strong correlation with the metrics calculated using the labels obtained from the semi-supervised segmentation algorithm. This evidences the quality of the algorithm in providing useful metrics to map the dataset into a feature space. We expect the VessMAP dataset to be useful for future studies regarding the influence of image and tissue characteristics on the generalization capability of segmentation algorithms.

## 4.2 Neural network performance on VessMAP splits

Many current methods for semantic segmentation of biological images involve neural networks [59]. Convolutional Neural Networks (CNNs) have been successfully used for segmenting biological structures for many years, especially after the adoption of encoder/decoder architectures, such as the original U-Net [60] and its variants [61–63]. Recently, the emergence of the Transformer [64] architecture allowed remarkable performance in tasks such as the development of Large Language Models [65,66], speech processing [67], multimodal learning [68], and drug discovery [69]. Regarding image processing, Transformers don't make assumptions about the relationship between the pixels of an image. This lack of inductive bias makes it harder to train a model from scratch in scenarios of scarce data, and it is usually necessary to pre-train a Vision Transformer (ViT) [70] in large datasets such as ImageNet [71]. Since we aim to evaluate the VessMAP performance using small training sets, we chose to use CNNs, which tend to perform better than ViTs for medical image segmentation with limited data [72].

To evaluate the potential of VessMAP to generate data splits that are challenging for neural networks, we generated eight different splits based on the features used for creating the dataset: blood vessel density, contrast, medial line heterogeneity, and noise estimation. For each feature, we selected 20 of the samples with the lowest and highest values and trained a segmentation CNN using two configurations: (i) training with samples that have the lowest feature values –lowest split– and evaluating with samples that have the highest feature values –highest split–, and (ii) training with the highest split and evaluating with the lowest split. We chose to use 20 images because it is a similar number to common split sizes used for well-known blood vessel datasets, such as DRIVE [13], STARE [14], and CHASEDB1 [73]. The idea behind this experiment is to test whether we can use VessMAP to generate splits that challenge the generalization capability of CNNs.

For this experiment, we used the CNN architecture illustrated in Fig 9. This architecture encodes the input data through a series of residual blocks [74], concatenates the resulted feature vector with the activations from the first convolution operation (similar to a U-Net [60]), and decodes the feature vector with a single residual block. For each training/evaluation split, we trained the network for 1000 epochs. The training was carried out using the Cross-Entropy as the loss function, the Adam optimizer [75], and a polynomial learning rate scheduler (power = 0.9)—which decays the initial learning rate (0.01) almost linearly.

Fig 10 presents the loss curves of the eight training setups (two split configurations for each metric). We evaluate the distance between the training and validation loss curves, $\delta$, as a metric of how well the CNN generalized for out-of-distribution data. Only the first 200 epochs are plotted because $\delta$ did not change significantly during the remaining epochs. We define $\delta$

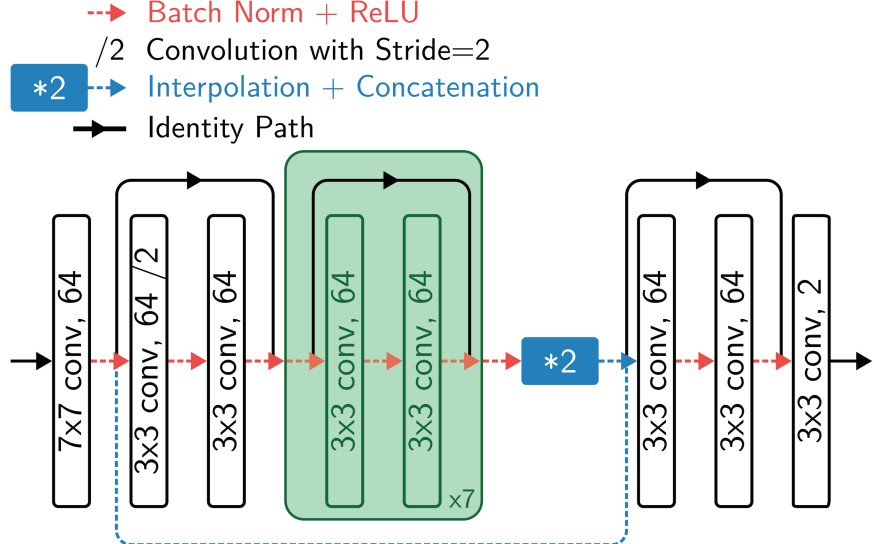

**Fig 9. CNN architecture used to evaluate the training performance of different VessMAP splits.** The input data go through a convolution layer with 64 7×7 kernels. 8 residual blocks are then applied, with the first one having stride 2 to downsample the activations. The final feature map of the encoder is concatenated with the activations from the first convolution layer and decoded by a single residual block. A final convolution layer generates the segmentation. The convolutions are padded to ensure no resolution loss after each operation.

as the difference between the training loss and the validation loss at a specific epoch. It is also worth noting that the loss curves were smoothed using an exponentially weighted moving average in order to reduce the natural variance of the loss values and obtain a more precise $\delta$ value. Here, we calculate $\delta$ at epoch 150.

For the splits using the contrast feature, when training with samples having low contrast (Fig 10b), the network generalizes well for new data having high contrast. This behavior can be attributed to the fact that high-contrast images are less challenging and, if the network learns how to properly segment low-contrast images, it tends to handle well high-contrast images. The opposite behavior occurs when we invert the training and validation sets. When training with high-contrast images (Fig 10a), the CNN could not generalize towards low-contrast data, yielding a negative $\delta$. The same behavior can be observed for the blood vessel density splits. Notice that a negative $\delta$ indicates that using specifically low-density samples as the training set yields low generalization towards more dense images. The exact opposite happens when training with denser samples. It can also be noticed that the training splits using the highest noise and medial line heterogeneity values resulted in similar training and validation loss curves. This indicates that, although these splits are unbalanced regarding feature values, the samples are still diverse enough to allow good generalization. The results of the experiments with different splits are summarized in Table 2.

Considering that the VessMAP images are diverse, another approach for generating challenging training and validation splits is to select samples that are far apart in the feature space. To do so, it is first necessary to identify a distance threshold above which the training and validation sets can be considered to be adequately separated in the feature space. We calculated this threshold by generating 10000 random splits of 20 training and validation images and obtaining the smallest Euclidean distance between all pairs of points of the two sets for each split. Then, we analyzed the histogram of the calculated distances and considered that two

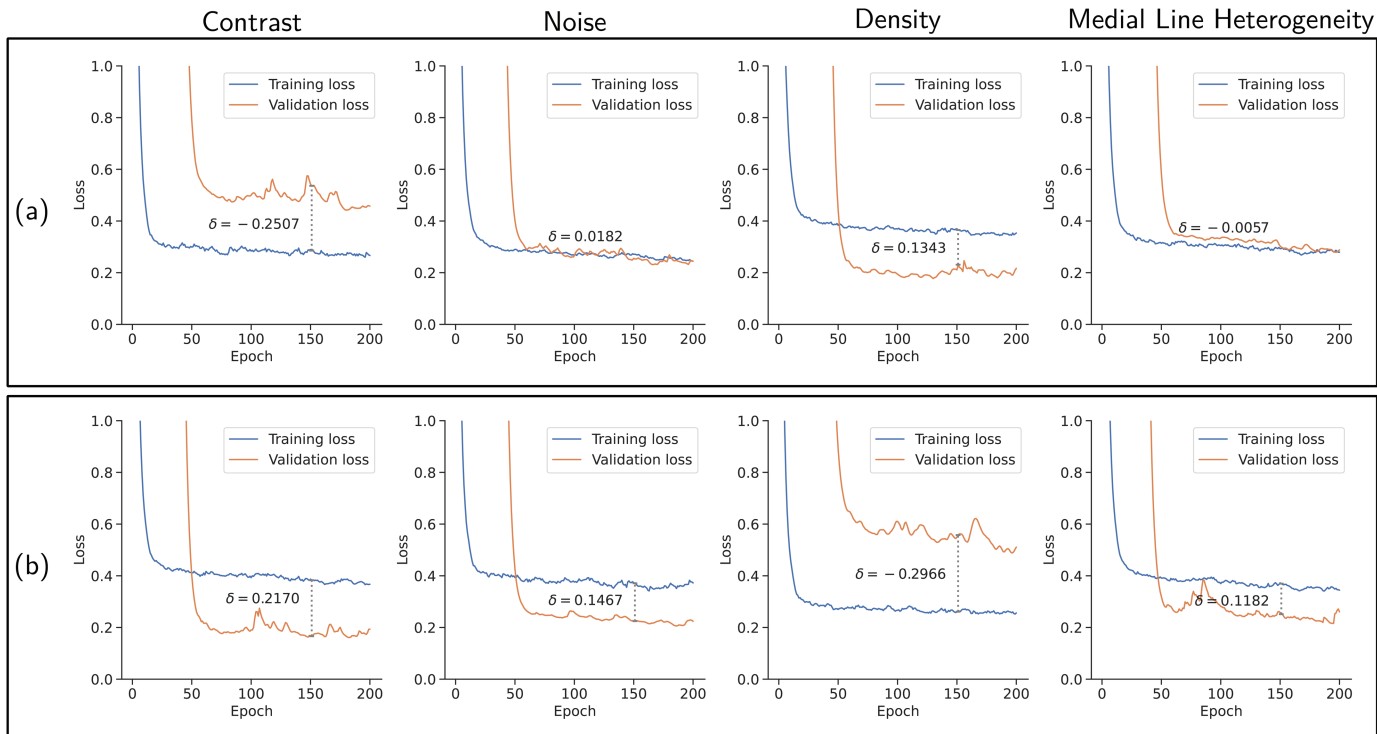

**Fig 10. Training and validation loss curves for eight different splits that were generated using the VessMAP metadata.** In (a), we depict the results obtained by training a CNN with the 20 samples having the highest values of each considered metric (indicated above each plot) and validating with the 20 images having the lowest values of the metric. The opposite situation is presented in (b), *i.e.*, the samples having the lowest values of the metrics were used for training, and the models were validated on the samples having the largest values of each metric. From left to right, each column shows the result for each considered metric: contrast, noise, density, and medial line heterogeneity. Negative $\delta$ values indicate that the CNN does not generalize well to the validation data. Positive $\delta$ values indicate that the validation samples are easier to segment than the training samples.

**Table 2. $\delta$ values for the experiments with different splits generated using the VessMAP metadata. Each experiment name depicts the feature used to split the dataset, followed by (h) if the training set had the largest values of the feature or (l) if the training set had the lowest values. Negative $\delta$ values suggest poor generalization and were marked as bold. *Skel. het: skeleton heterogeneity.**

| Experiment | Train loss | Valid loss | $\delta$ |
|---|---|---|---|
| **Contrast (h)** | **0.2853** | **0.5359** | **−0.2507** |
| Contrast (l) | 0.3831 | 0.1661 | 0.2107 |
| Density (h) | 0.3652 | 0.2309 | 0.1343 |
| **Density (l)** | **0.2610** | **0.5575** | **−0.2966** |
| Noise (h) | 0.2717 | 0.2535 | 0.0182 |
| Noise (l) | 0.3710 | 0.2243 | 0.1467 |
| **Skel. het. (h)** | **0.2934** | **0.2992** | **−0.0057** |
| Skel het. (l) | 0.3696 | 0.2514 | 0.1182 |

sets are far apart if their distance is larger than a threshold of $t = 0.7$, which corresponded to approximately 2.4% of the randomly drawn splits.

One of the identified splits containing highly distinct samples is highlighted in blue and red in Fig 8. By using the same CNN and hyperparameters as the previous experiments, an average validation Dice score [76] of $0.824 \pm 0.008$ was obtained for 100 training runs using the identified split. When the training and validation sets were swapped, a Dice score of $0.892 \pm 0.006$ was obtained. This result is in agreement with our previous experiment depicted

in Fig 10, as the images of the training set present high contrast, low density, and low medial line heterogeneity.

For reference, we ran a similar experiment on the retinography images from the DRIVE, STARE, and CHASEDB1 datasets. For the DRIVE dataset, a Dice of $0.803 \pm 0.002$ was obtained using the official split of the dataset, and a Dice of $0.791 \pm 0.003$ was obtained when the training and validation sets were swapped. Since the STARE dataset does not have an official training and validation split, we trained the network for 100 randomly drawn splits and calculated the average performance difference between each split and its swapped counterpart. An average Dice difference of $0.029 \pm 0.024$ was obtained, with a maximum difference of 0.12. This same approach was applied to the CHASEDB1 dataset, where an average Dice difference of $0.01 \pm 0.008$ was obtained, with a maximum difference of 0.03. Note that the CHASEDB1 and DRIVE datasets got similar results regarding the performance difference between splits. The difference in Dice values obtained for the splits of the VessMAP dataset, $0.892 - 0.824 = 0.068$, was significantly larger than the maximum values obtained for the CHASEDB1 and DRIVE datasets.

Interestingly, our experiments show that the STARE dataset contains the split with the largest performance difference among all datasets. Indeed, some samples of the STARE dataset have very distinct appearances when compared to the typical characteristics of the dataset. Thus, in addition to VessMAP, STARE also seems to be a suitable dataset for evaluating network generalizability on blood vessel segmentation tasks. Nevertheless, the higher number of images in VessMAP compared to STARE and the feature metadata enables the definition of training setups with a greater number of training/validation splits.

It is worth mentioning that the Dice values obtained in our experiments with the fundus images are slightly lower than the state-of-the-art results for these datasets [77]. This is mainly because no preprocessing and data augmentation were applied in order to match the training setup applied to VessMAP.

The heterogeneity of the VessMAP dataset is particularly useful for developing robust segmentation models when the training data is scarce. To show this, we ran a series of experiments using only four images for the training set. This replicates situations where, for instance, an active learning method suggests a small set of images for annotation, or on interactive segmentation scenarios where only a small set of blood vessels might be annotated. A neural network was trained on 4 randomly selected samples from the VessMAP dataset and the performance was measured on the remaining 96 samples. The same process was repeated 100 times using different sets of samples. To show that the overall results of our analyses are not dependent on a specific network architecture or training parameters, we replicated the same model and training approach used in [78]. Specifically, the $\phi_{3,8}$ U-Net model containing 6 convolution layers in the encoder was used. The training protocol was also replicated with the exception of the cyclical learning rate scheduler, which was replaced by a polynomial scheduler. The batch size was also changed from 4 to 2 since the training set has only four samples.

For each image of the dataset, we measured the Dice scores obtained for trainings runs in which the image was not included in the training set. The result for all images is shown in Fig 11a. It is clear that, for most images, the samples used for training the network have a large influence on the quality of the segmentation. The training set can lead to either very good segmentations or to segmentations that are of very poor quality.

For comparison, we repeated the same experiments for the DRIVE, STARE, and CHASEDB1 datasets. The results are shown in Fig 11b–11d. The variation observed for these datasets is much smaller compared to the VessMAP dataset. That is, four training samples are usually enough to obtain good and robust performance on the remaining samples. Thus,

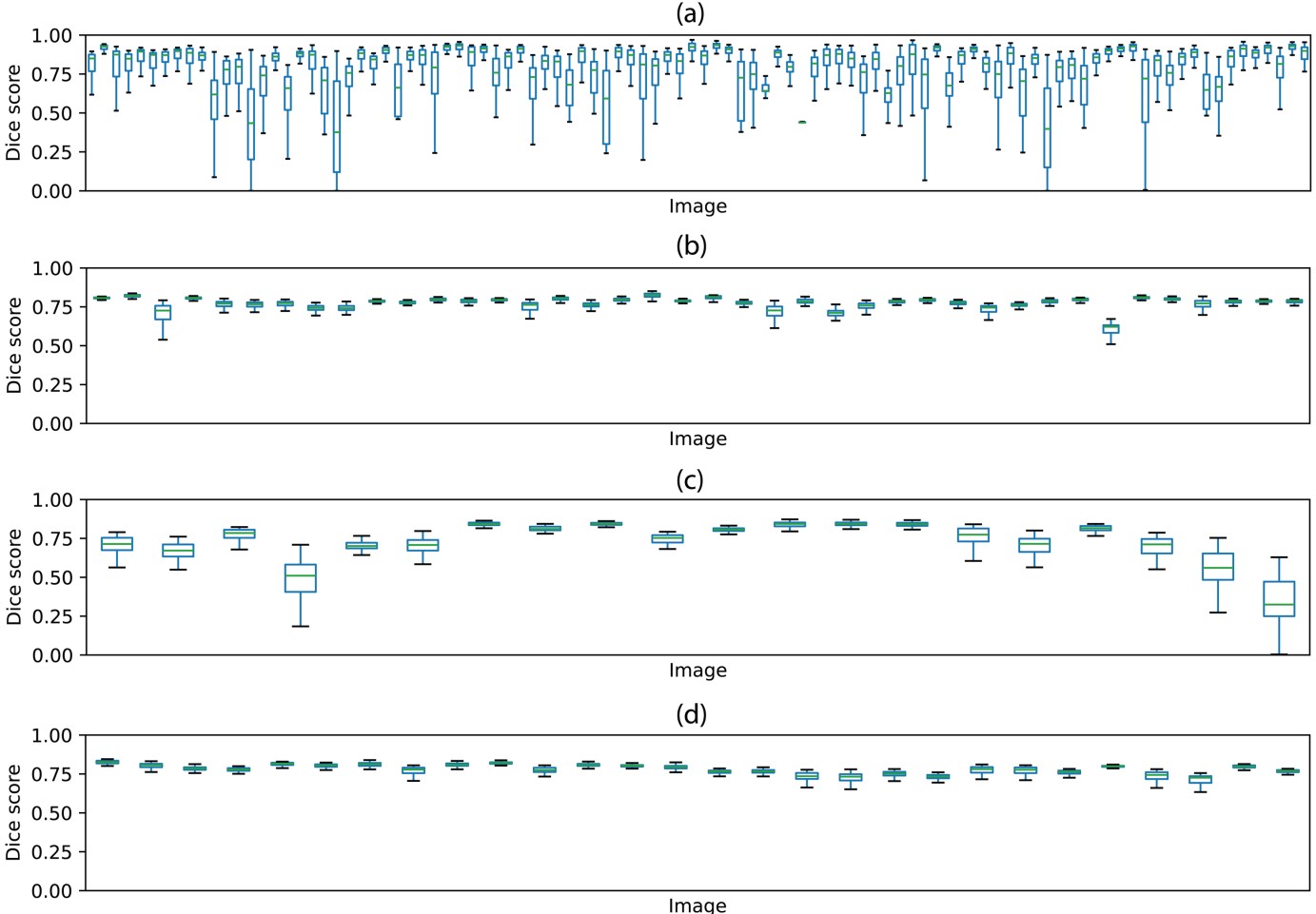

**Fig 11. Segmentation performance for all images of the datasets used in the experiments.** Each vertical box represents the distribution of Dice scores obtained for a sample across 100 training runs. The bottom and top of each box represent, respectively, the first ($q_1$) and third ($q_3$) quartiles of the data. The horizontal green line indicates the median and the whiskers indicate the range $[q_1 - 1.5(q_3 - q_1), q_3 + 1.5(q_3 - q_1)]$. The results are shown for the (a) VessMAP, (b) DRIVE, (c) STARE, and (d) CHASEDB1 datasets.

methods developed to work on scarce data annotation regimes might trivially result in good, low-biased performance when tested on these datasets. The same trend was observed for the area under the ROC curve (AUC) and average precision performance metrics (S1 Fig and S2 Fig of the supporting information).

To quantify the performance variation observed, the difference between the highest and lowest Dice score obtained for each sample was calculated. The average difference for all samples was then calculated for each dataset. The values are shown in Table 3 and confirm the high performance variations observed for VessMAP.

In Fig 12 we show example segmentations obtained for the sample having the median standard deviation of Dice scores among all runs, that is, a sample with a typical variation of Dice scores observed in the dataset. The training set can lead to many missing blood vessels, to an oversegmentation of the vessels, to the presence of spurious holes as well as to discontinuities on the vessels. For many other samples, we also observed a large number of false positives.

**Table 3. Influence of the training set on the generalizability of a model.** For each dataset, models were trained on 100 different training sets containing 4 images each and were validated on the remaining images. The difference between the maximum and minimum Dice scores obtained for each sample across all runs was calculated and averaged over all samples. The standard deviation of the differences is also shown to provide a reference regarding the degree of variation observed among samples.

| Dataset | Average Dice difference | Std. dev. Dice difference |
| --- | --- | --- |
| VessMAP | 0.55 | 0.19 |
| DRIVE | 0.13 | 0.13 |
| STARE | 0.29 | 0.18 |
| CHASEDB1 | 0.11 | 0.06 |

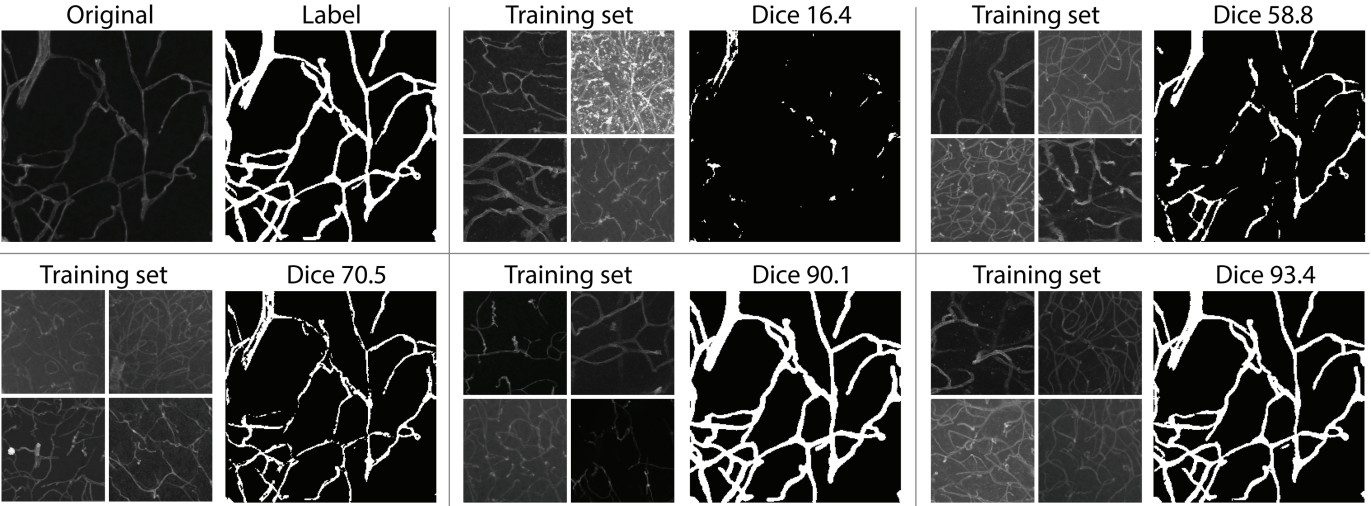

**Fig 12. Example segmentations obtained for sample 6818 of the VessMAP dataset.** The original image and manual annotation are shown at the upper left corner. The remaining panels show the training set used to train a neural network and the respective segmentation obtained for the sample. The respective Dice scores are shown above each segmentation.

With these analyses, we suggest two applications of the VessMAP metadata. First, one can generate splits that challenge the generalization capacity of a neural network, yielding negative $\delta$ during training. This kind of split can be used to test or develop new approaches to handle datasets having very distinct samples. In a similar fashion, splits that have positive $\delta$ can be used for developing new active learning methods, where it is useful to identify challenging samples for training networks so as to obtain low validation loss. In both situations, an ideal model should converge the training and validation loss curves, resulting in $\delta \approx 0$.

To aid the development of such methods, we provide official splits of the dataset containing training sets that lead to vastly distinct inference performances. The splits are shown in Table 4. For calculating the performance of each split, each of the 100 training runs was repeated 5 times using different seeds for the random number generator used during training.

## 5 Conclusion

Annotating appropriate images from a larger dataset for training machine learning algorithms is an important task. This is because the usual approach is to use as many images as possible. While this approach is relevant for general classification problems, for medical image segmentation, where image annotation can be very costly, the images used must be carefully

**Table 4. Relevant training splits of the VessMAP dataset. Each row shows the average Dice score obtained on the remaining 96 samples when training a neural network using the four images indicated in the training set column. The standard deviation obtained across five repetitions of the training runs is also shown.**

|         | Training set                  | Dice score         |
|---------|-------------------------------|--------------------|
| Split 1 | 4404, 11828, 16295, 7344      | 0.846 ± 0.007      |
| Split 2 | 12943, 8493, 12618, 9284      | 0.800 ± 0.014      |
| Split 3 | 7083, 6887, 14778, 2287       | 0.752 ± 0.025      |
| Split 4 | 12877, 15577, 12960, 9593     | 0.702 ± 0.024      |
| Split 5 | 8284, 9284, 11411, 9452       | 0.653 ± 0.008      |
| Split 6 | 9710, 2643, 11111, 8196       | 0.589 ± 0.073      |

selected in order to ensure good coverage of different tissue appearances and imaging variations. In addition, it is important that the annotated images do not lead to biases in downstream tasks related to tissue characterization. For instance, training segmentation algorithms mostly on prototypical images can lead to incorrect measurements on samples having unusual properties (e.g., very bright or very noisy).

We used an intuitive sampling methodology that evenly selects, as best as possible, both typical and atypical vascular image samples for creating VessMAP, a dataset containing a heterogeneous set of samples representing many possible variations of image noise and contrast as well as blood vessel density and intensity variance. One important characteristic of the dataset is that it provides an intuitive uniform grid in the feature space that can be used for further analyses. For example, one can study the accuracy of a segmentation model on different regions of the grid to identify regions where samples are not being correctly segmented. A robust algorithm should provide good segmentation no matter if a sample is too noisy, bright or dark, if it has low or high contrast, or any other variation on relevant image properties. The dataset is being made available together with the metadata containing the features used for creating the dataset.

We showed that different splits of the dataset can lead to largely distinct validation performances. The heterogeneity is particularly noticeable when training data is scarce. For many popular blood vessel datasets, the vasculature has similar characteristics throughout all samples. Thus, while they can be used for testing novel approaches for segmenting blood vessels, they are not ideal for quantifying the robustness of methods under small distribution shifts regarding sample characteristics and vessel geometry. Our analyses showed that VessMAP displays stronger appearance changes, with an average Dice score change of 0.55, depending on the samples used for training. This result contrasts with the average Dice score difference of 0.29 observed for the STARE dataset, the most heterogeneous dataset identified in the experiments after VessMAP.

One drawback of VessMAP is that the samples are relatively easy to segment. When training with more than 20 samples, the validation performance tends to be good and has little dependence on the training set. Thus, the usefulness of the dataset lies mostly in tasks with very limited training data. Another important consideration is that the features used to create the dataset are not necessarily related to the underlying conditions affecting the tissue samples (e.g., wild type, mutations, stroke, development stage) or to the acquisition process of the samples. Thus, obtaining good performance on the VessMAP dataset is important but not sufficient to conclude that a model is not biased on downstream tasks.

We expect that the dataset will be useful for studies regarding data distribution shifts as well as few-shot, interactive segmentation and active learning methods. We suggest two specific applications. Observing the official splits shown in Table 4, it is clear that among the 100 samples, training on samples 4404, 11828, 16295, and 7344 (split 1) led to robust mod-

els displaying good segmentation accuracy on the remaining samples (Dice score of 0.846). The same is not true for most of the other samples in the dataset. An active learning method should be able to automatically identify these four samples since they lead to a very low annotation effort to segment the whole dataset with good accuracy.

Another interesting application is the automatic identification of segmentation mistakes. A common scenario in real applications is the following. A new dataset is provided and needs to be segmented for downstream analyses. Since manually annotating blood vessel samples is time-consuming, only a fraction of the samples are manually annotated. A segmentation model is then trained on the annotated samples and applied to the remaining images. But how do we verify that the annotated samples were enough to provide good accuracy on downstream analyses for the remaining data? Looking back at Table 4, if the manually annotated samples are those of split 6, the performance of the model is known to be poor (Dice score of 0.589). Thus, one can develop additional heuristics to identify where the model is making mistakes. For instance, an interesting prospect is to analyze the topology of the vasculature and automatically identify missing segments, spurious branches and unrealistic connectivity patterns. The official splits of the VessMAP dataset allow a systematic comparison between methods developed by different research groups.

Interestingly, the splits in Table 4 represent different degrees of difficult for such methods. Split 6, with a Dice score of 0.589, should lead to clearly unrealistic connectivity patterns. However, the difference between splits 1 and 2 is likely more subtle, and automatically identifying segmentation mistakes in split 2 that are not on split 1 should be more challenging.

The VessMAP dataset might also be used for testing the performance of more general methods that were not developed specifically for segmenting blood vessels. Many large-scale biomedical datasets have been created in recent years [79–83]. Fluorescence microscopy samples are relatively uncommon in such datasets. Thus, the VessMAP dataset can be useful as an additional imaging modality for quantifying the performance of general methods.

## Supporting information

**S1 Fig. Segmentation performance for all images of the datasets used in the experiments.** Each vertical box represents the distribution of the area under the ROC curve obtained for a sample across 100 training runs. The bottom and top of each box represent, respectively, the first ($q_1$) and third ($q_3$) quartiles of the data. The horizontal green line indicates the median and the whiskers indicate the range $[q_1 - 1.5(q_3 - q_1), q_3 + 1.5(q_3 - q_1)]$. The results are shown for the (a) VessMAP, (b) DRIVE, (c) STARE, and (d) CHASEDB1 datasets.
(PDF)

**S2 Fig. Segmentation performance for all images of the datasets used in the experiments.** Each vertical box represents the average precision obtained for a sample across 100 training runs. The values were calculated as the average precision obtained when setting the decision threshold to each unique probability value. The bottom and top of each box represent, respectively, the first ($q_1$) and third ($q_3$) quartiles of the data. The horizontal green line indicates the median and the whiskers indicate the range $[q_1 - 1.5(q_3 - q_1), q_3 + 1.5(q_3 - q_1)]$. The results are shown for the (a) VessMAP, (b) DRIVE, (c) STARE, and (d) CHASEDB1 datasets.
(PDF)

## Author contributions

**Conceptualization:** Cesar H. Comin.

**Data curation:** Matheus Viana da Silva, Natália de Carvalho Santos.

**Formal analysis:** Matheus Viana da Silva.

**Investigation:** Matheus Viana da Silva.

**Methodology:** Matheus Viana da Silva.

**Project administration:** Cesar H. Comin.

**Resources:** Julie Ouellette, Baptiste Lacoste.

**Software:** Matheus Viana da Silva, Natália de Carvalho Santos.

**Supervision:** Cesar H. Comin.

**Validation:** Matheus Viana da Silva.

**Visualization:** Matheus Viana da Silva.

**Writing – original draft:** Matheus Viana da Silva, Natália de Carvalho Santos, Cesar H. Comin.

**Writing – review & editing:** Julie Ouellette, Baptiste Lacoste.

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
