## [Decision Letter · Decision Letter 0]

30 Dec 2024

PONE-D-24-51696A new dataset for measuring the performance of blood vessel segmentation methods under distribution shiftsPLOS ONE

Dear Dr. da Silva,

Thank you for submitting your manuscript to PLOS ONE. After careful consideration, we feel that it has merit but does not fully meet PLOS ONE’s publication criteria as it currently stands. Therefore, we invite you to submit a revised version of the manuscript that addresses the points raised during the review process.

While preparing your manuscript, please make sure that you address the concerns raised by the reviewers related to the biases your proposed method may have introduced in the dataset generaton method. Moreover, please include further statistical analysis metrics in your findings. The architectural details of the CNN used in this study need to be reported in detail for reproduction of these results. Furthermore, the choice of various parameters in the considered CNN have to be justified and compared against others in an ablation study to make sure no inductive bias has been introduced. 

We look forward to receiving your revised manuscript.

Kind regards,

Muhammad Bilal, Ph.D.

Academic Editor

PLOS ONE

Journal Requirements:

“Cesar H. Comin thanks FAPESP (grant no. 21/12354-8) for financial support. M. V. da Silva thanks FAPESP (grant no. 23/03975-4), Google's Latin America Research Awards (LARA 2021), and the Google PhD Fellowship Program for financial support. The authors acknowledge the support of the Government of Canada's New Frontiers in Research Fund (NFRF) (NFRFE-2019-00641).”

Reviewers' comments:

Reviewer's Responses to Questions

**Comments to the Author**

1. Is the manuscript technically sound, and do the data support the conclusions?

Reviewer #1: Yes

Reviewer #2: Yes

Reviewer #3: Yes

2. Has the statistical analysis been performed appropriately and rigorously? 

Reviewer #1: Yes

Reviewer #2: Yes

Reviewer #3: Yes

3. Have the authors made all data underlying the findings in their manuscript fully available?

Reviewer #1: Yes

Reviewer #2: No

Reviewer #3: Yes

4. Is the manuscript presented in an intelligible fashion and written in standard English?

Reviewer #1: Yes

Reviewer #2: Yes

Reviewer #3: Yes

5. Review Comments to the Author

Reviewer #1: 1. Abstract

- Recommendation: Rewrite the abstract to:

- Highlight the practical applications of the proposed dataset and methods.

- Explicitly mention the novel aspects of the research.

- Include a brief summary of the key results.

---

2. Introduction

- Recommendation:

- Expand the introduction to discuss how the methodology could be applicable to other areas, such as autonomous driving, aerial imaging, or manufacturing.

- Clearly articulate the specific gap in the literature that this research addresses.

---

3. Methodology

- Recommendation:

- Include a flowchart or diagram to visually represent the dataset creation process and evaluation framework.

- Provide more examples or illustrations of how the dataset was curated to ensure heterogeneity.

---

4. Results

- Recommendation:

- Include a deeper analysis of the results, focusing on the strengths and weaknesses of the proposed approach.

- Compare the proposed method to additional state-of-the-art techniques to strengthen the validity of the conclusions.

---

5. Practical Applications

- Recommendation:

- Add a section or paragraph that explores real-world scenarios where this research could have a significant impact (e.g., radiology diagnostics, AI-assisted surgery).

- Highlight the potential for this methodology to improve robustness in AI applications outside of healthcare.

---

6. Figures and Tables

- Recommendation:

- Add captions that provide more context and make the figures more standalone.

- Include qualitative results, such as segmented images from the dataset, to complement the quantitative metrics.

---

7. Discussion

- Recommendation:

- Elaborate on the broader significance of the findings for machine learning and computer vision.

- Discuss potential limitations, such as computational requirements or biases in the dataset, and suggest ways to address them in future work.

---

8. Conclusion

- Recommendation:

- Highlight how the dataset can serve as a benchmark for future research in segmentation under distribution shifts.

- Suggest specific avenues for extending the work, such as applying the methodology to larger or more diverse datasets.

---

9. Language and Readability

- Recommendation:

- Revise for grammatical correctness and conciseness.

- Use simpler language for technical terms to improve accessibility for readers outside the immediate field.

---

10. References

- Recommendation:

- Add citations from the latest studies (post-2021) to strengthen the relevance and contextualization of the research.

Reviewer #2: 1. In the abstract, the authors should mention the achieved results and the model's improvement over existing approaches.

2. In the related work section, various old papers are added. I suggested adding the recent relevant literature from 2022- 2024 to provide an updated overview to the readers.

3. The paper's main contributions of the paper should be mentioned in points at the end of the introduction section.

4. I did not find any table representing the results of the CNN model.

5. In the deep learning section, the authors are advised to add necessary introduction related to deep learning and its importance by citing the recent deep learning models such as iAFPs-Mv-BiTCN, AIPs-DeepEnC-GA, DeepAVP-TPPred, PAtbP-EnC, Deepstacked-AVPs, and AIPs-SnTCN for the reader's concern to provide a broader overview.

6. An up to date comparison of the proposed model with existing state of the art models will be necessary.

7. What should be the future directions of the proposed model

Reviewer #3: The manuscript titled "A New Dataset for Measuring the Performance of Blood Vessel Segmentation Methods Under Distribution Shifts" introduces the VessMAP dataset, a highly heterogeneous collection of annotated fluorescence microscopy images. The dataset aims to improve the evaluation of segmentation algorithms, particularly under challenging conditions like distribution shifts.

1. The sampling methodology ensures diversity, but how does it account for potential biases introduced by the original dataset's composition? Could the over-representation of certain conditions affect downstream analyses?

2. You mention dual annotations for 20 samples. What metrics (e.g., Cohen's kappa, Dice similarity) were used to assess inter-annotator variability, and how do these findings support the reliability of your annotations?

3. The features used (e.g., contrast, noise) are domain-specific. Could these features inadvertently bias model evaluations? Have you explored alternative or additional features that might better generalize to unseen data?

6. PLOS authors have the option to publish the peer review history of their article (what does this mean?). If published, this will include your full peer review and any attached files.

Reviewer #1: No

Reviewer #2: No

Reviewer #3: No

---

## [Author Response · Author response to Decision Letter 1]

15 Feb 2025

We provided a complete response for the editor and reviewers as a PDF file on submission.

---

## [Decision Letter · Decision Letter 1]

16 Mar 2025

A new dataset for measuring the performance of blood vessel segmentation methods under distribution shifts

PONE-D-24-51696R1

Dear Dr. da Silva,

We’re pleased to inform you that your manuscript has been judged scientifically suitable for publication and will be formally accepted for publication once it meets all outstanding technical requirements.

Kind regards,

Muhammad Bilal, Ph.D.

Academic Editor

PLOS ONE

Additional Editor Comments (optional):

Reviewers' comments:

Reviewer's Responses to Questions

**Comments to the Author**

1. If the authors have adequately addressed your comments raised in a previous round of review and you feel that this manuscript is now acceptable for publication, you may indicate that here to bypass the “Comments to the Author” section, enter your conflict of interest statement in the “Confidential to Editor” section, and submit your "Accept" recommendation.

Reviewer #1: All comments have been addressed

Reviewer #2: All comments have been addressed

2. Is the manuscript technically sound, and do the data support the conclusions?

Reviewer #1: Yes

Reviewer #2: Yes

3. Has the statistical analysis been performed appropriately and rigorously? 

Reviewer #1: I Don't Know

Reviewer #2: Yes

4. Have the authors made all data underlying the findings in their manuscript fully available?

Reviewer #1: Yes

Reviewer #2: Yes

5. Is the manuscript presented in an intelligible fashion and written in standard English?

Reviewer #1: Yes

Reviewer #2: Yes

6. Review Comments to the Author

Reviewer #1: Authors have made requested changes. Authors responds to each of suggestion and try to explain the asked quires.

Reviewer #2: The required comments are successfully incorporated and paper is significantly improved. No further comments from my side

7. PLOS authors have the option to publish the peer review history of their article (what does this mean?). If published, this will include your full peer review and any attached files.

Reviewer #1: No

Reviewer #2: No

---

## [Editor Report · Acceptance letter]

PONE-D-24-51696R1

PLOS ONE

Dear Dr. da Silva,

I'm pleased to inform you that your manuscript has been deemed suitable for publication in PLOS ONE. Congratulations! Your manuscript is now being handed over to our production team.

Kind regards,

on behalf of

Dr. Muhammad Bilal

Academic Editor

PLOS ONE